# Spatial evolution, influencing factors and spillover effects of logistics resilience in the Yangtze River Economic Belt

**Xiaofan Zhang**, **Yin Huang** *

School of Logistics, Central South University of Forestry and Technology, Changsha, Hunan, China

* share0122@126.com

**Data Availability Statement:** All relevant data are within the manuscript and its Supporting Information files. The data underlying the results presented in the study are available from the China Urban Statistical Yearbook.

## Abstract

Logistics resilience is a significant representation of sustainable development ability and a necessary support for high-quality economic development. In order to explore the influencing factors and realization mechanism of the improvement of logistics resilience of the Yangtze River Economic Belt and the high-quality and sustainable development of the economy, this paper comprehensively considers factors of the supply and demand relationship of the logistics market, industrial structure and ecological environment, and evaluates the urban logistics resilience of the Yangtze River Economic Belt by using POI data and statistical data. Combined with the spatial Durbin model, the influencing factors and spatial spillover effects of inter-city logistics resilience were revealed. This study found that the urban logistics resilience in the lower reaches of the Yangtze River has been high. Except Chongqing and Shanghai, the COVID-19 epidemic happened in 2020 led to a significant decrease in logistics resilience. In the meanwhile, every 1% increase in the logistics resilience of the city will promote the logistics resilience of the adjacent cities by 0.145%. Economic condition and urban development potential have positive effects on logistics resilience of the city and its adjacent cities. The economic condition has a direct effect coefficient of 0.166 and an indirect effect coefficient of 0.181, The direct and indirect effects of urban development potential are significantly positive, and the coefficients are 0.001 and 0.006, respectively. The level of information, government support and ability of technological innovation are helpful for the improvement of urban logistics resilience while hindering the enhancement of logistics resilience in adjacent cities. The research area can be extended in the future and more influencing factors can be considered in the future.

## Introduction

After the COVID-19 pandemic, the inherent vulnerability and weak resilience of the logistics system have been gradually exposed, and the mobility of the global production network has been cut off. It is urgent to establish a more stable and resilient logistics system [1]. In the post epidemic era, the just ended COVID-19 has had a tremendous impact on economic and social development [1,2]. As an important component of the supply chain, the logistics system has

**Funding:** This study was supported by the National Natural Science Foundation of China (Grant no.72174214, to Yin Huang); Key Project of Scientific Research of Department of Education of Hunan Province (Grant no. 21A0156, to Yin Huang); and the Hunan Provincial Innovation Foundation for Postgraduate (Grant no. CX20220737, to Xiaofan Zhang). The funders had no role in study design, data collection and analysis, decision to publish, or preparation of the manuscript.

**Competing interests:** The authors have declared that no competing interests exist.

gradually exposed its inherent vulnerability and weak resilience [1], and the normal operation of the logistics industry has been strongly and continuously affected [2–4]. In terms of short-term effects, the main manifestation is the cut off of liquidity, and a series of measures such as restricting population mobility and traffic control taken in response to the epidemic have caused logistics delays, leading to material shortages [3,4]. In the long run, the cumulative effects of repeated outbreaks of the epidemic have led to multiple pressures such as tight human resources, traffic control, rising logistics costs, and delayed transmission of the industrial chain, making it difficult for the logistics industry to recover [5], thus becoming the main obstacle for other industries to restore normal production order. Given its core position in the supply chain, the logistics industry has become a key link in global economic recovery [4,5], and its resilience characteristics and influencing factors have also received increasing attention.

The term resilience has evolved from "resilio" in the Arabic language and has gone through three stages: engineering resilience, ecological resilience and evolutionary resilience [6]. In recent years, the concept of resilience has gradually extended to supply chain [7], logistics [8], tourism [9] and other fields. Logistics resilience aims to understand the ability of logistics system to respond to disasters, hazards, disturbances and changes [8]. Urban logistics resilience can measure the resistance of urban logistics to emergencies and potential as well as driving forces for sustainable development of regional logistics, which is a key facilitator of resource optimization, economic development and sustainable environmental construction within cities and regions [10].

The Yangtze River Economic Belt is one of the most important river basin economic belts in China, spanning nine provinces and two municipalities directly under the central government, playing an important role in promoting the complementary advantages of the upper, middle and lower reaches of the Yangtze River, and is the key force for the optimization of regional economic pattern and high-quality development [11,12]. The radiation area of the Yangtze River Economic Belt is broad [13], and there are obvious differences in developmental foundation and industrial structure between different regions [14], leading to imbalanced development. As the integration of regional construction and development in the Yangtze River Economic Belt has progressed [14], issues like excessive resource concentration in logistics and market segmentation brought on by this phenomenon have become more prominent and have seriously hampered the improvement of logistics resilience and overall sustainable development of cities in the Yangtze River Economic Belt [13]. In this context, what are the spatial distribution characteristics of logistics resilience in cities within the Yangtze River Economic Belt? What factors will affect the improvement of urban logistics resilience? What is the interaction relationship between logistics resilience among different cities? These are all the contents that this study will explore.

In light of this, we construct an urban logistics resilience evaluation system that includes four aspects: logistics supply, logistics demand, industrial structure, and the impact of logistics on the ecological environment. We calculate the logistics resilience of cities in the Yangtze River Economic Belt from 2016 to 2020 and analyze the laws and characteristics of their spatial and temporal evolution. To empirically analyze the influencing factors and spatial spillover effects of logistics resilience, we use a spatial econometric model. From a new perspective of logistics resilience, it can provide practical basis for regional sustainability and high-level development of the Yangtze River Economic Belt.

The rest of this article is arranged as follows: Chapter 2 is a literature review, mainly introducing relevant literature on logistics resilience, sustainable development of the Yangtze River Economic Belt, and their relationship. Chapter 3 introduces the data sources and research methods used in this article, including the construction of the evaluation system of logistics

resilience and the selection process of spatial econometric models. Chapter 4 presents the calculation results based on the evaluation system of logistics resilience and spatial Durbin model in Chapter 3. Chapter 4 includes three parts: logistics resilience calculation, identification of influencing factors, and calculation of spatial spillover effects. Chapter 5 discusses the innovation of this article and proposes relevant suggestions based on the research results. Chapter 6 summarizes the main conclusions and limitations of this article, and illustrates the parts that can be improved in the future research.

## Literature review

### Study on resilience and logistics resilience

Resilience was originally used to measure the ability of metals to recover after deformation under external forces, and is widely used in the field of physics. Holling first introduced the concept of resilience in ecology in the 1970s to measure and describe the state characteristics of ecosystems [15]. Since then, as an important concept and indicator to examine the ability of different regions and industries to flexibly respond to emergencies and maintain normal operations, resilience has been gradually introduced by domestic and foreign scholars from the field of ecology to urban planning, disaster prevention, disaster reduction and other multidisciplinary fields. Resilience has gradually become an important scientific concept that has received attention in areas such as regional development and urban planning, following the concepts of sustainable development and green development [16]. However, although the concept of resilience has gradually been widely applied, due to differences in research background and research issues, the definition of resilience has not yet been unified. In general, resilience is often defined as the ability of a system, society, or region to respond to, adapt to, and recover from sudden, disruptive events quickly and effectively [17,18]. From an evolutionary perspective, resilience can be understood as the ability of an organization or system to continuously restructure and evolve to become more complete and efficient. With the deepening of research, scholars have generally recognized that its concept is not only "maintaining stability" and "restoring the original state" covered by "engineering resilience" and "ecological resilience", but also "updating, transforming, establishing a new growth path and resisting risks" emphasized by adaptive resilience based on the evolutionary perspective. In the definition of resilience. Martin *et al.* added the concepts of vulnerability, resistance, robustness and resilience, as well as the four processes of resistance, recovery, reorientation and path reconstruction [19,20]. Based on the analysis of resilience and adaptability of global production networks, theories of resilience are constructed in evolutionary economic geography [21].

Most existing researches on resilience focus on urban economic resilience [22,23], transportation resilience [24,25], infrastructure resilience [26,27], supply chain resilience [28,29], etc.. In the field of logistics and supply chain, some scholars have carried out research on the definition, measurement methods and influencing factors of supply chain resilience. Bak *et al.* [30], after reviewing 101 literatures, defined the supply chain resilience of small and medium-size enterprise (smes) as the combination of enterprise adaptation, response ability, sustainability and competitiveness, reflecting the ability of enterprises to maintain the status quo, enter a new ideal state or self-renewal. Sezer *et al.* [31] analyzed the influencing factors of supply chain resilience and the importance levels of different factors, and found that the most important factor for improving supply chain resilience was the adaptive ability of enterprises.

As an important link in the supply chain, the definition of logistics resilience tends to be confused with that of supply chain resilience (as shown in Table 1). Studies on its resilience mainly focus on the resilience of logistics enterprises. For example, Maharjan and Kato [32] analyzed the impact of COVID-19 on corporate logistics and supply chain activities through

Table 1. Definitions of supply chain resilience and logistics resilience.

| Scholars and time | Definition |
|---|---|
| Christopher & Peck (2004) [34] | The ability of the supply chain to return to normal operation or even improve its current state after external disruptions. |
| Gaonkar (2007) [35] | The ability to recover and resume normal operation after a system outage. |
| Falasca (2008) [36] | The ability of the supply chain to mitigate the likelihood and impact of disruption, including the time required for the supply chain to return to normal capacity. |
| Ponomarov & Holcomb (2009) [37] | The adaptive ability to prepare for unexpected events, the control ability to respond to unexpected events and maintain the original status, and the ability to recover the original structure and function after an unexpected event. |
| Pettit, Fiksel & Croxton (2010) [38] | The ability of complex logistics systems to continue to exist (survive), adapt, and grow in the face of external disturbances or changes. |
| Barroso (2011) [39] | In order to maintain the supply chain objectives to cope with the adverse impact of emergencies on the supply chain ability. |
| Jones et al.(2014) [40] | The ability of a supply chain system to recover to its original state within a certain period of time after disruption. |
| Purvis et al. (2016) [41] | Resilience reflects the ability of an organization or business to prepare for disruptions that may come from customers, suppliers, or related business processes. |
| Zaczyk & Liebert (2020) [42] | The ability of the logistics system to continue to meet the social logistics needs and reduce the possibility of interference and its negative impact in the face of the volatile changes in the environment. |

semi-structured interviews with Japanese logistics companies. Ketudat and Jeenanunta [33] used the same methodology to study the key factors behind the high logistics resilience of three Thai logistics companies during the COVID-19 pandemic.

In the study of economics and management, supply chain resilience has always been an important topic. The current research mainly analyzes the measurement [43], shaping [44] and optimization [45] of the toughness of supply chain management from the whole process. As an important link in the supply chain, logistics resilience can reflect the ability of different entities in the logistics industry to withstand shocks and recover from the impact of business cycle and external shocks [8]. Regional logistics is a subsystem of regional resilience, developing in a spiral cycle. While improving its own resilience, the logistics industry can also have a significant promoting effect on the development of regional economic resilience [46]. Therefore, the regional logistics resilience is defined in this paper as the resistance of the regional logistics system when it is disturbed by external economic, natural and other environmental conditions, or the recovery and improvement ability of the regional logistics system to turn to a better path through system optimization. The development process of regional logistics resilience is a spiral upward process, and the disturbance of external conditions cannot be completely understood as a negative impact. In the process of adapting to the impact and finding a better path, regional logistics will comprehensively improve its self-capability and play a role as an important help for regional economic resilience.

At present, the academic research on logistics resilience focuses on the evaluation of logistics resilience [47], the analysis of spatial evolution mechanism [48], and the exploration of influencing factors [49]. Based on the theory of regional economic resilience, Yu et al. [50] conducted a study on the spatio-temporal heterogeneity of logistics timeliness from the perspective of resilience. Jin et al. [47] constructed a logistics resilience evaluation system from four aspects: logistics economic capacity, socio-demographic capacity, community connection capacity and innovation capacity, taking into account not only economic and social environmental factors, but also logistics facilities, demand and innovation capacity. Xie et al. [51] adopted analytic Hierarchy Process (AHP) combined with fuzzy comprehensive evaluation,

selected evaluation indicators from the economic environment, demand level, government support level, scientific and technological development level and other perspectives to build an aviation logistics resilience evaluation system, and diagnosed the overall weak resilience of China's aviation logistics after the epidemic and its causes. However, in addition to logistics demand, economic development level, social conditions, innovation ability and other factors, few studies have incorporated ecological environmental factors into the logistics resilience evaluation system. Logistics and transportation activities are one of the main sources of environmental pollution. Considering the impact of logistics activities on the ecological environment has a certain significance for the evaluation and improvement of logistics resilience. In addition, the research on the influencing factors of logistics resilience is relatively scarce, and the analysis of its spatial spillover effect is also insufficient. In the context of economic recovery, differences in logistics resilience may bring about changes in regional development patterns. How to accurately understand and measure the characteristics of logistics resilience, describe the spatial pattern and time changes of logistics resilience, and identify the key factors affecting logistics resilience are of great significance to the economic recovery and sustainable development in the post-epidemic era, as well as to expand the research scope and theoretical depth of resilience and logistics geography.

## Study on logistics industry and logistics resilience in the Yangtze River Economic Belt

The Yangtze River Economic Belt is one of the most important river economic belts of regional development in China. It plays an important role in expanding domestic demand and optimizing economic development pattern in the process of regional development in China. However, in the process of rapid industrialization and urbanization after the reform and opening up, the Yangtze River Economic Belt is faced with such problems as excessive population growth, deterioration of ecological environment, extreme consumption of resources, backward industrial structure, and imbalance of regional development [52]. These problems have reduced the sustainable capacity of urban development to varying degrees.

In recent years, relevant studies on the development of the Yangtze River Economic Belt mainly focus on urban impact evaluation [53], urban cluster study [54], industrial undertaking and transfer discussion [55], regional connection and cooperation [56], etc. Studies on the regional logistics of the Yangtze River basin and the sustainable development of the Yangtze River Economic Belt are also limited to the Yangtze River Delta [57,58]. However, the tertiary industry still accounts for the largest proportion in the industrial structure of the Yangtze River Economic Belt, and the tertiary industry, with the logistics industry as the pillar industry, promotes the rapid economic development [59]. However, affected by factors such as the level of logistics infrastructure, logistics industry layout and urban location, the spatial differentiation of regional logistics development in the Yangtze River Economic Belt is obvious [60]. It is found that there was significant regional heterogeneity in the development level of urban logistics industry along the Yangtze River Economic Belt, with the eastern region leading, the central region rising and the western region lagging behind [61]. And the logistics competitiveness of the Yangtze River Economic Belt presents a spatial pattern of "the downstream coastal area is the strongest, and the middle and upper inland areas are cross-distributed" [62]. However, no matter in the relatively developed eastern region or the relatively backward western region, the changes between the development of logistics industry and economic growth are in the same direction [63]. In other words, the development of regional logistics plays a positive role in promoting regional economic growth. Therefore, a comprehensive understanding of the spatial pattern of the logistics industry and in-depth disclosure of

its resilience characteristics and influencing factors are of great significance for exploring the upgrading path of the development of the logistics industry and promoting the sustainable development of the Yangtze River Economic Belt.

The impact of logistics industry on the sustainable development of the Yangtze River Economic Belt is mainly reflected in the population flow, infrastructure construction and ecological environment. Gong and Zhu [64] have found that developing low-carbon logistics can help rationally allocate urban functions and reduce its negative impact on the environment. Shee et al. [65] empirically studied the impact of smart logistics on the environmental, social and economic dimensions of urban sustainable development, and found that the development of smart logistics is conducive to the improvement of smart city environment, which in turn is conducive to the improvement of social and economic performance. Cao [66] pointed out that problems such as waste gas, noise and traffic congestion generated by the logistics industry would have a negative impact on the urban environment. Therefore, an effective mechanism should be established to evaluate the sustainability of the urban logistics industry. Resilience reflects the ability of something to resist disruption and continue to evolve. Logistics resilience provides a powerful tool for urban sustainability research of the Yangtze River Economic Belt.

Logistics resilience is a branch of the study of resilience, which can reflect the sustainable development ability of logistics industry in the Yangtze River Economic Belt from the perspective of logistics facility construction and logistics service level. The Yangtze River Economic Belt has a vast radiation area, and different regions differ greatly in resource endowment, industrial layout, infrastructure and development stage, thus leading to significant internal development imbalance. As an intermediate link connecting production, circulation and consumption, logistics contributes to the efficient allocation of resources within a region and plays an important role in promoting industrial structure adjustment, industrial upgrading and regional sustainable development [32,34]. Therefore, it is of great significance to combine the study of logistics resilience with the sustainable development of the Yangtze River Economic Belt and analyze the influencing factors and spatial spillover effects of the sustainable development of the logistics industry for promoting the sustainable development of the Yangtze River Economic Belt.

Current studies on the development of logistics in the Yangtze River Economic Belt mainly focus on the coupling and coordination between logistics and other industries or ecological environment. Ye et al. [67] comprehensively considered the coupling relationship among logistics industry, urbanization and ecological environment in the Yangtze River Economic Belt, and used entropy weight method and coupling coordination degree model to describe the spatio-temporal evolution model of the coupling coordination among the three. Gong et al. [68] investigated the sustainable development ability and coordinated development level of the manufacturing and logistics industries in the Yangtze River Economic Belt from both time and space dimensions by constructing a three-stage super-efficiency SBM model. Du and Yang [69] summarized the overall coordination degree and development law of logistics and manufacturing industries in the Yangtze River Economic Belt by constructing the Haken model, and determined the order of their coordinated development. However, studies on sustainable development of the Yangtze River Economic Belt from the perspective of logistics resilience are relatively scarce. Therefore, this paper studies the spatial and temporal pattern, influencing factors and spatial spillover effects of logistics resilience in the Yangtze River Economic Belt, identifies problems in the process of urban sustainable development from the perspective of logistics industry construction, and then proposes countermeasures and suggestions for the improvement of logistics resilience and sustainable construction in the Yangtze River Economic Belt.

## A summary of the literature review

In general, domestic and foreign scholars have defined, measured, identified and optimized supply chain and logistics resilience from various perspectives such as enterprise management, logistics and supply chain management. These theoretical and empirical studies provide scientific reference for the analysis of spatial evolution, influencing factors and spillover effects of logistics resilience, and are also the basis for this study, but there are still some deficiencies.

First, most of the current research on resilience focuses on economic resilience, infrastructure resilience, supply chain resilience, etc., while there are few studies on logistics resilience and no clear and appropriate definition of the concept of logistics resilience. Second, current studies on the resilience of logistics and supply chain mainly focus on the fields of operation research optimization, logistics and supply chain management. Most studies construct evaluation systems to measure and compare the resilience of logistics in research units, and explore the influencing factors leading to the differences in the resilience of logistics in different units, but do not take into account the role of spatial spillover effect caused by geographical proximity. It is urgent to study from the perspective of economic geography. Third, the importance of the logistics industry to the sustainable development of the Yangtze River Economic Belt is self-evident, but the current research on the logistics industry and the sustainable development of the Yangtze River Economic Belt mainly focuses on the coupling and coordination between logistics and other industries or ecological environment, and the research from the perspective of logistics resilience needs to be in-depth and improved.

Therefore, from the perspective of resilience, this paper intends to measure the logistics resilience and its evolution law of cities in the Yangtze River Economic Belt from four aspects: logistics supply, logistics demand, industrial structure and impact on the environment, and analyze its influencing factors and the spatial spillover effect between cities. From the perspective of logistics industry construction, this paper identifies the problems existing in the process of sustainable development of economy, industry and ecological environment of the city and the Yangtze River Economic Belt, and puts forward suggestions to improve the logistics resilience and sustainable construction of the Yangtze River Economic Belt. The framework of this study is shown in Fig 1.

## Methodology

### Logistics resilience evaluation framework

The development status and potential of urban logistics based on the development situation of regional logistics in the Yangtze River Economic Belt must be fully taken into consideration in order to reveal the influencing factors and spillover effects of urban logistics resilience. This paper evaluates logistics resilience in the Yangtze River Economic Belt from four main aspects, namely logistics supply, logistics demand, industrial structure, and impact on environment. The indicators included in each dimension are shown in Table 2.

### Logistics supply

Affected by the COVID-19 epidemic, in addition to the rising human resource costs and transportation costs due to the tight supply and demand of personnel, rising oil prices and other factors, logistics enterprises also need to bear additional epidemic prevention and control costs, so the overall logistics costs are increasing. The increase of logistics costs, the closure of logistics enterprises, the reduction of logistics employees, the interruption of transportation and other logistics service providers have reduced, and the resulting imbalance between supply and demand in the logistics market has threatened the logistics resilience. Natural resources,

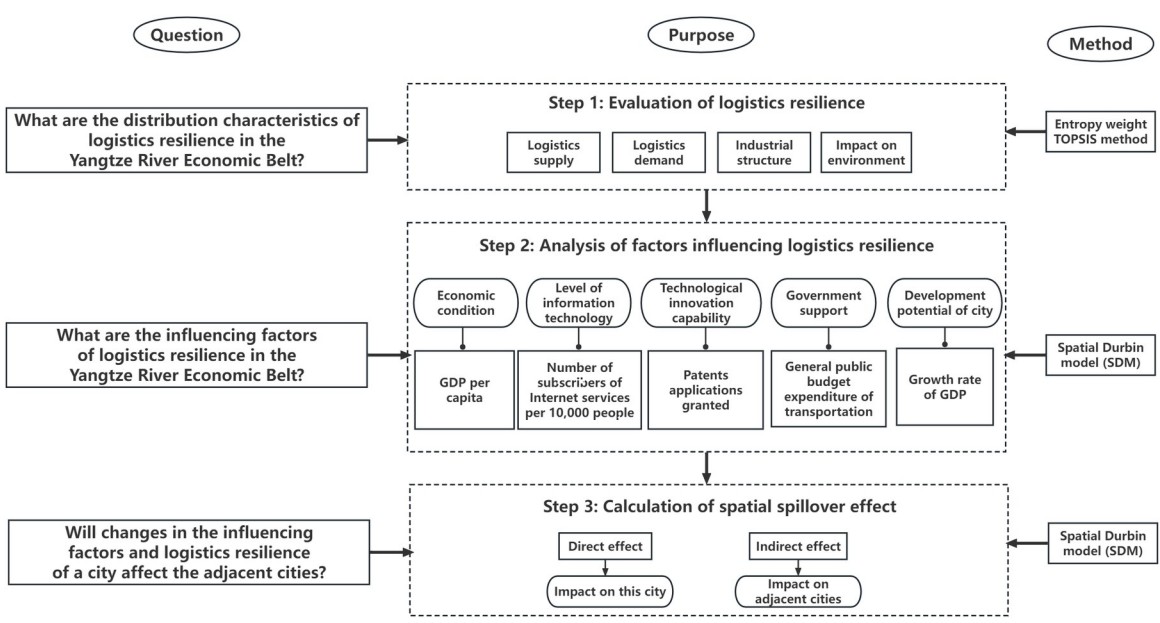

**Fig 1. Research framework.**

human resources, technical capability, and capital are just a few examples of the production factors that are directly connected to the growth of the logistics industry [13]. From the perspectives of infrastructure, capital support, and human resources, we choose three variables: highway mileage, fixed asset investment in transportation, and the number of employees in

**Table 2. Logistics resilience evaluation system.**

| Aspects | Indicators | Data source | Remarks |
|---|---|---|---|
| Logistics supply | Highway mileage | Statistical yearbook | |
| | Fixed asset investment in transportation | Statistical yearbook | |
| | The number of employees in the post and Telecommunications industry | Statistical yearbook | |
| Logistics demand | Population density | Statistical yearbook | It's the number of people divided by the total area |
| | Industrial output above scale | Statistical yearbook | |
| | Total retail sales of social consumer goods | Statistical yearbook | |
| Industrial structure | The added value of the primary industry | Statistical yearbook | |
| | The added value of the secondary industry | Statistical yearbook | |
| | The added value of the wholesale and retail industry | Statistical yearbook | |
| Impact on environment | The total freight volume | Statistical yearbook | |
| | The number of logistics enterprises | POI data set | |
| | The number of trucks | Statistical yearbook | |

the post and telecommunications industry. These variables are combined with the development characteristics of the logistics industry in the Yangtze River Economic Belt. Production factors such as facilities, labor and capital are the basis for the sustainable development of the logistics industry, and the three metrics selected in this paper can show how resilient logistics are in terms of production factors.

## Logistics demand

One of the main evaluation indexes of logistics resilience is whether the relationship between supply and demand is balanced, that is, whether the existing logistics development foundation can support market demand. Affected by the COVID-19 epidemic, on the one hand, the demand such as e-commerce terminal distribution has soared, and on the other hand, due to the obstruction of accessibility, logistics enterprises cannot work normally. The supply side of the logistics market cannot meet the growing market demand, and the fluctuation of demand and the imbalance between supply and demand have brought great challenges to the maintenance and improvement of logistics resilience. Therefore, the population density, industrial output above scale, and total retail sales of social consumer goods are selected to evaluate the logistics demand. From the standpoint of domestic and overseas circulation, the total amount of imports and exports can reflect the social demand for the logistics sector. The main segment of the logistics market demand is industrial logistics, and the industrial output above scale reflects the potential logistics demand and industry vitality from a production standpoint. The scale of consumer goods logistics is continually expanding due to commercial activity, and total retail sales of social consumer goods reflect the industrial demand for logistics services, particularly the express delivery industry from a consumer standpoint [13]. The demand of logistics industry mainly comes from the production, circulation and other economic activities of other industries. The continuous demand is an important guarantee for the sustainable development of regional logistics.

## Industrial structure

Industrial structure evaluates logistics resilience from the perspective of external environment. Logistics industry is a producer service industry, which is greatly affected by the change of industrial structure. Affected by the COVID-19 epidemic, the logistics demand structure in the fields of industry, import, people's livelihood and regeneration is constantly changing, and the trend of the promotion of "double cycle" and the iterative upgrading of manufacturing industry will also promote industry segmentation and iterative upgrading. These changes will have a certain impact on the toughness of logistics. To measure the effect of industrial structure on logistics resilience, the added value of the primary industry, the added value of the secondary industry, and the added value of the wholesale and retail industry are selected to reflect the industrial structure. Compared with other tertiary industries, the logistics industry has strong inclusiveness, derivative and dynamic, which is attached to the real industry and is derived from production, circulation and consumption activities. In the current socio-economic system, the logistics industry is closely linked with other industries, especially the physical manufacturing industry. In other words, the demand for logistics industry comes from other industrial production, trade and other economic activities. The coordination and integration among industries can promote the integration and cooperation of regional development, thus contributing to sustainable development. Therefore, the resilience of the logistics industry can be evaluated by analyzing the growth and rationality of structural of the three major industries.

## Impact on environment

This paper selects the total freight volume, the number of logistics enterprises and the number of trucks to evaluate the logistics resilience of cities from the perspective of ecological environment. Freight is one of the functions of logistics, which is often realized through highway transportation, railway transportation and air transportation. In these processes, transportation tools will produce a large number of dust, noise and tail gas pollution, which is the main source of environmental pollution. Controlling energy consumption and pollution can reduce environmental damage and contribute to the recycling of energy and other factors, thus promoting the sustainable development of the logistics industry. Therefore, environmental protection is not only the future developmental needs of the logistics industry, but also an important part of sustainable development. The ecological environment is the foundation for the long-term sustainable development of regional industry and the guarantee for improving the logistics resilience.

## Research area and data source

**Research area.** The Yangtze River Economic Belt is a belt that traverses eastern and western hinterlands and carries on the northern and southern regions in China. The Yangtze River Economic Belt covers 11 provinces and municipalities, including Shanghai, Anhui, Zhejiang, Jiangsu, Jiangxi, Hubei, Hunan, Sichuan, Chongqing, Yunnan and Guizhou, and contains 130 cities (including ethnic minority autonomous prefectures). Considering the feasibility and convenience of data acquisition, this paper takes 110 prefecture-level cities in the Yangtze River Economic Belt (excluding ethnic minority autonomous prefectures) as the research units. The specific research area is shown in Fig 2.

## Data source

1. Data from statistical yearbook
   In order to assess the logistics resilience of the Yangtze River Economic Belt and calculate its influencing factors and spatial spillover effects, this paper comprehensively selected panel data samples from 110 prefecture-level cities (excluding ethnic autonomous prefectures) from 2016 to 2020. In the variable setting process of the logistics resilience evaluation system and its influencing factors, except for population density and the number of logistics enterprises, the remaining data were derived from the statistical yearbooks and bulletins of each city from 2016 to 2020, and the linear interpolation method was used to fill in the missing data. Population density can be obtained by quadratic calculation, which is numerically equal to the ratio of the total urban population to the land area of the administrative region. Both the number of urban population and the land area of the administrative region used for calculation are from the statistical yearbook.

2. POI data
   In the logistics resilience evaluation system, the "number of logistics enterprises" data comes from the point of interest (POI) data set. The POI data comes from the API of Auto-Navi open platform (https://lbs.amap.com). The POI data set contains geographical location information of logistics enterprises in cities of the Yangtze River Economic Belt. The specific acquisition method is to count the number of logistics enterprises in each city from 2016 to 2020 with the keywords of "logistics", "express", "express" and "freight" under the entry of "companies".

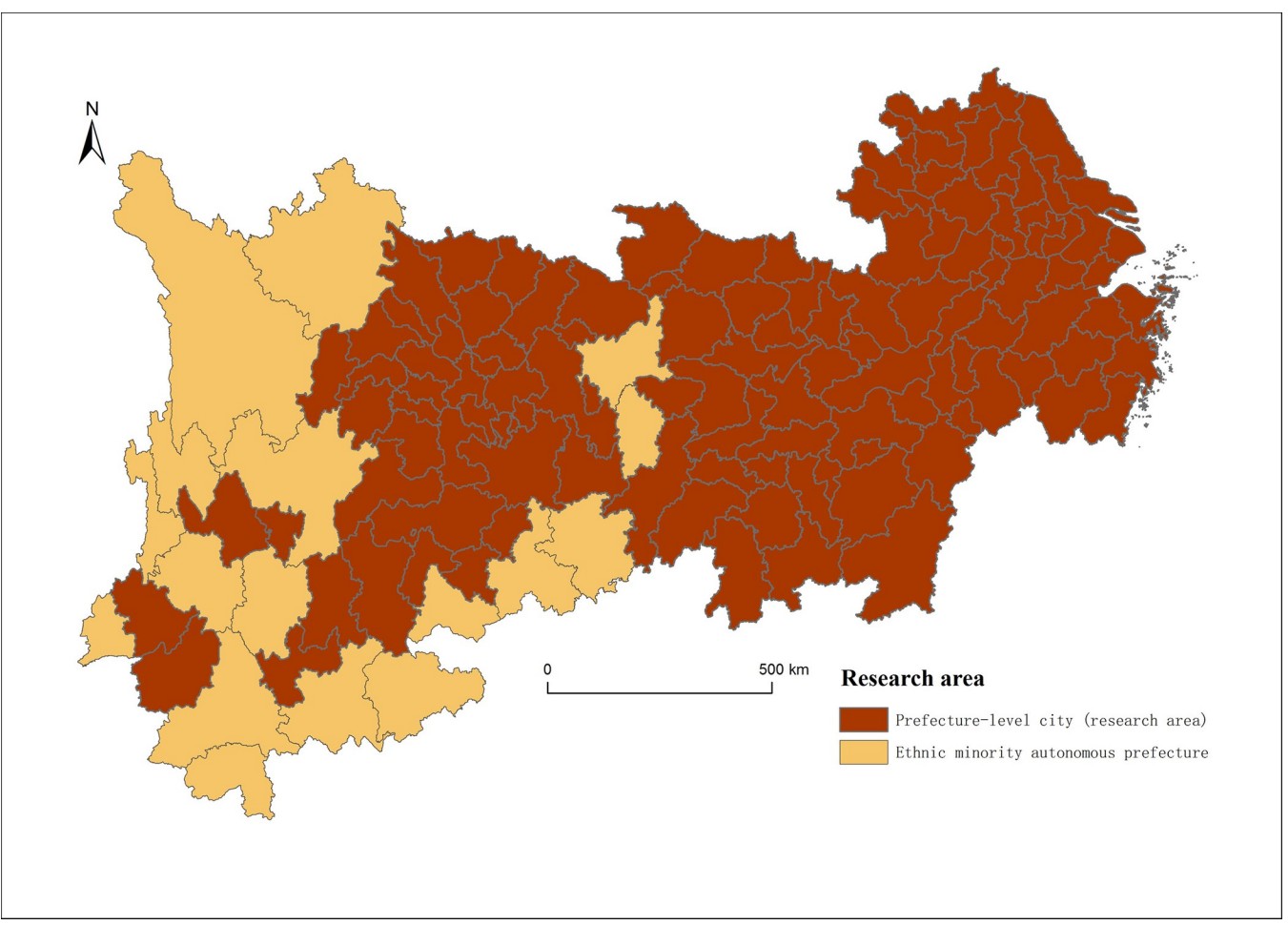

**Fig 2. Research area.** Republished from [70] under a CC BY license, with permission from [Resource and Environment Science and Data Center], original copyright [2023]. Fig 2 is similar but not identical to the original image and is therefore for illustrative purposes only.

## Method

### Entropy weight TOPSIS method

The entropy weight method obtains the weight of each index by calculating the information entropy. Technique for Order Preference by Similarity to an Ideal Solution (TOPSIS) reflects the gap between evaluation schemes by calculating the closeness between evaluation objects and optimal schemes [71,72]. Entropy weight TOPSIS method integrates the advantages of entropy weight method and TOPSIS method, and avoids the influence of subjective weighting on the analysis results [73]. The formula of entropy weight is given by:

$$\mathrm{H}(M) = -k \sum_{i=1}^{n} P(M_i) \times \ln P(M_i) \tag{1}$$

Where $k$ is a coefficient larger than 0, $0 < P(M_i) < 1$ and $\sum_{i=1}^{n} P(M_i) = 1$.

### Global spatial autocorrelation analysis

Before using the spatial measurement method, we should first test whether the sample has spatial dependence. We use the global Moran's $I$ index to characterize the spatial characteristics of

the global logistics resilience of the Yangtze River Economic Belt. The formula of global Moran's $I$ index is:

$$Global\ Moran's\ I = \frac{\sum_{i=1}^{n} \sum_{j\neq 1}^{n} W_{ij} Z_i Z_j}{\sigma^2 \sum_{i=1}^{n} \sum_{j\neq 1}^{n} W_{ij}} \tag{2}$$

$$Z_i = \frac{x_i - \bar{x}}{\sigma}, \sigma = \frac{1}{n} \sum_{i=1}^{n} (x_i - \bar{x})^2 \tag{3}$$

where n is the number of units; $W_{ij}$ is a space weight matrix; $x_i$ is the input-output ratio in region i; $Z_i$ is the standard transformation of $x_i$.

## Spatial econometric model

According to the geographical proximity effect, the spatial interaction between cities cannot be disregarded when analyzing the logistics resilience of the Yangtze River Economic Belt [74]. The conventional regression approach is not without flaws. On the basis of this, we introduce spatial factors and build a spatial econometric model to discuss the driving forces and spatial spillover effects of urban logistics resilience.

1. Space weight matrix setting
   Because the promotion of logistics resilience is affected by the difference of regional economic level and geographical distance, the spatial economic distance matrix considering the characteristics of spatial geographical distance and economic related attributes is selected as the spatial weight matrix. The specific expressions are as follows:

$$C_{ij} = (1/|PGDP_j - PGDP_i + 1|) \times e^{-d_{ij}} \tag{4}$$

   Among them, $C_{ij}$ is the spatial economic distance matrix, $PGDP_j$ and $PGDP_i$ represent the per capita GDP of cities j and cities I respectively, d is the geographical distance.

2. Model setting
   The spatial lag model (SAR), spatial error model (SEM), and spatial Durbin model (SDM) are currently the most popular spatial econometric models. In order to achieve the best fitting results, SAR, SEM, and SDM models are constructed, taking into account the various economic implications revealed by the various types of spatial econometric models. The models are set as follows:

1. Spatial lag model (SAR)
   The SAR model primarily examines the spatial dependence of explained variable to determine whether there is a spatial spillover effect in a particular region [74]. The level of development of the logistics sector in neighboring cities will have an impact on the growth of the urban logistics sector. In order to investigate the spatial spillover effect of urban logistics resilience in the Yangtze River Economic Belt, a SAR model was developed.

$$LC_{i,t} = \rho w_{ij} LC_{j,t} + \beta_1 lnECO_{i,t} + \beta_2 lnINF_{i,t} + \beta_3 lnTEC_{i,t} + \beta_4 lnGOV_{i,t} + \beta_5 DEV_{i,t} + \varepsilon_{i,t} \tag{5}$$

2. Spatial error model (SEM)
   SEM model mainly reflects the relationship between variables through the spatial dependence of disturbance error. In addition to the economic condition, technological

innovation capability and other factors selected in this paper, logistics resilience will also be affected by unpredictable factors such as geographical location. Therefore, the SEM model is constructed to quantify the impact of this unobservable factor on logistics resilience.

$$\text{LC}_{i,t} = \beta_1 lnECO_{i,t} + \beta_2 lnINF_{i,t} + \beta_3 lnTEC_{i,t} + \beta_4 lnGOV_{i,t} + \beta_5 DEV_{i,t} + \mu_{i,t} \tag{6}$$

$$\mu_{i,t} = \lambda w_{ij}\mu_{j,t} + \varepsilon_{i,t}, \varepsilon_{i,t} \sim N(0, \delta^2 I_n) \tag{7}$$

3. Spatial Durbin Model (SDM)

SDM model is a combination of SAR model and SEM model. In order to measure the spatial interaction effect of logistics resilience of Yangtze River Economic Belt, and analyze the impact of different factors on the improvement of urban logistics resilience, the SDM model is constructed.

$$
\begin{aligned}
\text{LC}_{i,t} = {} & \rho w_{ij}\text{LC}_{j,t} + \beta_1 lnECO_{i,t} + \beta_2 lnINF_{i,t} + \beta_3 lnTEC_{i,t} + \beta_4 lnGOV_{i,t} + \beta_5 DEV_{i,t} \\
& + \theta_1 w_{ij}lnECO_{i,t} + \theta_2 w_{ij}lnINF_{i,t} + \theta_3 w_{ij}lnTEC_{i,t} + \theta_4 w_{ij}lnGOV_{i,t} + \theta_5 w_{ij}DEV_{i,t} \\
& + \varepsilon_{i,t}
\end{aligned}
\tag{8}
$$

In the formula, LCi, t is the explained variable, indicating the competitiveness of urban logistics. ECO represents the economic condition, INF represents the level of information, TEC represents technological innovation capability, GOV represents government support, and DEV represents urban development potential. $\beta_1$~$\beta_5$ are the regression coefficients of each explanatory variable, $w_{ij}$ is the spatial weight matrix, $\rho$ is the spatial autocorrelation coefficient of the explained variable, $\lambda$ is the spatial autocorrelation coefficient of the error term. $\theta_1$~$\theta_5$ denotes the spatial lag regression coefficient of each explanatory variable, $\varepsilon$ is a random error term.

## Results

### Evaluation and spatial characteristics of urban logistics resilience

**Evaluation of logistics resilience and analysis of spatio-temporal evolution characteristics.** The entropy TOPSIS method is used to calculate the comprehensive score of logistics resilience of 110 cities in the Yangtze River Economic Belt from 2016 to 2020 (Table 3), and ArcGIS software is used for geographical visualization (Fig 3).

According to the measured results of logistics resilience (Table 3 and Fig 3), affected by the epidemic, the logistics resilience of cities in the Yangtze River Economic Belt in 2020 decreased significantly (the color of most cities in Fig 1 became lighter), and the overall level was much lower than that of previous years (Table 3). However, the logistics resilience of Shanghai and Chongqing increased rather than decreased: compared with 2019, the logistics resilience of Shanghai increased from 0.829 to 0.876 in 2020, and that of Chongqing increased from 0.554 to 0.714. At the same time, the gap between Shanghai and Chongqing and other cities is widening. From 2016 to 2020, the logistic resilience of Chongqing has always been higher than that of most cities (Fig 3). The logistic resilience of Shanghai and Chongqing has been maintained above 0.520, while that of Suzhou, Wuhan, Wuxi, Ningbo and other cities is slightly lower than that of Shanghai and Chongqing. However, in 2020, the logistic resilience of Yangzhou, which ranks third, is only 0.128. It was far lower than Chongqing (0.714), which ranked second.

**Table 3. Logistics resilience from 2016 to 2020 (Top 5 and the last 5).**

|  | Top 5 |  | Last 5 |  |
|---|---|---|---|---|
| 2016 | Shanghai | 0.843 | Huangshan | 0.016 |
|  | Suzhou | 0.647 | Yaan | 0.013 |
|  | Chongqing | 0.640 | Anshun | 0.009 |
|  | Wuhan | 0.394 | Zhangjiajie | 0.006 |
|  | Chengdu | 0.391 | Lijiang | 0.001 |
| 2017 | Shanghai | 0.872 | Huangshan | 0.014 |
|  | Suzhou | 0.654 | Yaan | 0.010 |
|  | Chongqing | 0.567 | Anshun | 0.009 |
|  | Wuhan | 0.394 | Zhangjiajie | 0.005 |
|  | Wuxi | 0.369 | Lijiang | 0.001 |
| 2018 | Shanghai | 0.843 | Huangshan | 0.009 |
|  | Suzhou | 0.631 | Chizhou | 0.008 |
|  | Chongqing | 0.530 | Puer | 0.007 |
|  | Ningbo | 0.419 | Zhangjiajie | 0.006 |
|  | Wuhan | 0.404 | Lijiang | 0.001 |
| 2019 | Shanghai | 0.829 | Chizhou | 0.010 |
|  | Suzhou | 0.619 | Puer | 0.010 |
|  | Chongqing | 0.554 | Ziyang | 0.009 |
|  | Wuxi | 0.469 | Zhangjiajie | 0.003 |
|  | Wuhan | 0.443 | Lijiang | 0.002 |
| 2020 | Shanghai | 0.876 | Puer | 0.00023 |
|  | Chongqing | 0.714 | Ziyang | 0.00022 |
|  | Yangzhou | 0.128 | Yaan | 0.00020 |
|  | Yancheng | 0.112 | Lijiang | 0.00012 |
|  | Suqian | 0.067 | Zhangjiajie | 0.00001 |

From a regional perspective, the urban logistics resilience of the Yangtze River Delta urban agglomeration represented by Nanjing, Ningbo and Suzhou is relatively high (Fig 3). These cities are regional and even national logistics hubs, with advantages such as geographical location,

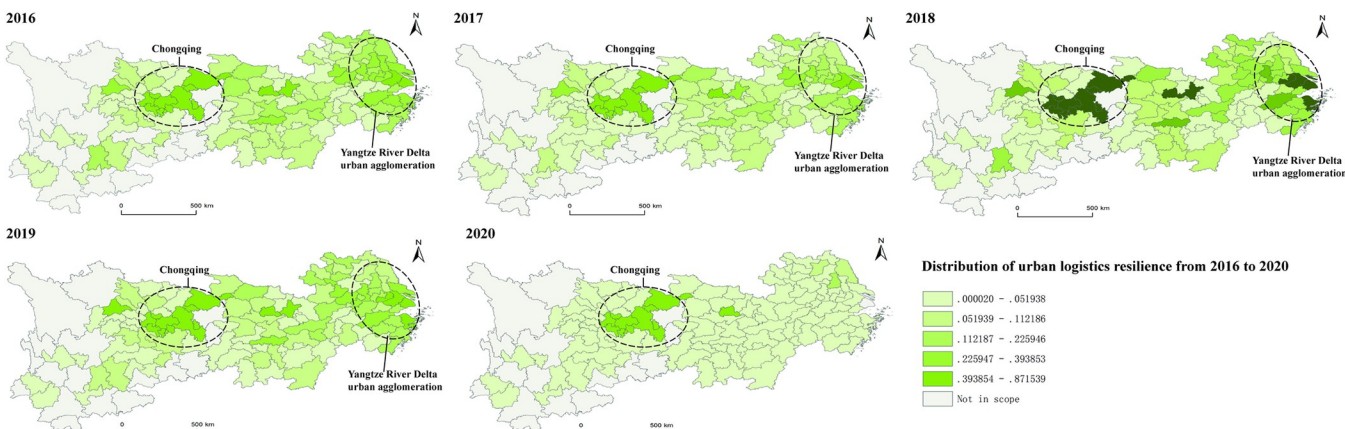

**Fig 3. Spatial-temporal evolution of logistics resilience in the Yangtze River Economic Belt.** Republished from [70] under a CC BY license, with permission from [Resource and Environment Science and Data Center], original copyright [2023]. Fig 3 is similar but not identical to the original image and is therefore for illustrative purposes only.

**Table 4. The global Moran's *I* index of logistics resilience from 2016 to 2020.**

| Year | Moran's *I* | Z value | P value |
|------|-------------|---------|---------|
| 2016 | 0.267 | 1.427 | 0.014 |
| 2017 | 0.254 | 1.241 | 0.015 |
| 2018 | 0.271 | 1.524 | 0.017 |
| 2019 | 0.259 | 1.27 | 0.024 |
| 2020 | 0.222 | 1.352 | 0.122 |

developed economy, preferential policies and talent concentration. As a result, these cities have always been relatively resilient in logistics and can still exert their competitive advantages during the COVID-19 pandemic to lead the sustainable development of regional logistics. Among the cities in the middle reaches of the Yangtze River, except for Wuhan in 2018, the logistics resilience of other cities from 2016 to 2020 is mostly stable and always at the lower level. This phenomenon also indicates that the middle reaches of the Yangtze River lacks the leadership and drive of core cities. When the epidemic broke out in 2020, the logistics resilience of only one city, Wuhan, was 0.0074, while that of other cities including Changsha and Nanchang, the two provincial capitals, was all less than 0.0038, indicating that there is still a large room for improvement. However, the logistics resilience of most cities in the upper reaches of the Yangtze River (such as Lijiang, Puer and Zhangjiajie in Table 2) has been low before and after the epidemic (Fig 3).

**Spatial agglomeration characteristics of logistics resilience.** The spatial correlation test of the urban logistics resilience in the Yangtze River Economic Belt is conducted to ascertain whether there is self-correlation prior to using the spatial econometric model to assess the spatial effect of various factors on logistics resilience. The calculation results of global Moran's *I* index are shown in Table 4.

The findings demonstrate that the global Moran's *I* index of logistics resilience for 110 cities in the Yangtze River Economic Belt from 2016 to 2019 is significantly positive. This finding suggests that there is a certain spatial positive correlation between logistics resilience of cities, that is, cities with similar logistics resilience will cluster together.

The global Moran's *I* index of logistics resilience in 2020 does not pass the test of visibility, but it does not mean that there is no spatial interaction effect between them and the adjacent areas. This phenomenon indicates that the spatial autocorrelation is limited to local areas, or the positive and negative spatial autocorrelations are offset each other [75]. Affected by the epidemic situation, the normal production of each city is affected, and the logistics resilience is significantly reduced. Strict prevention and control measures also interfere with the originally close exchanges and cooperation between cities and regions. Therefore, the logistics resilience of each city in 2020 does not show obvious agglomeration characteristics.

## Influencing factors of urban logistics resilience

**Model construction.**

1. Variable description

The logistics resilience calculated above is used as explained variables to analyze the influencing factors and spatial spillover effects of Yangtze River Economic Belt logistics resilience. Five factors are chosen as explanatory variables, including the economic condition, technological innovation capability, informational level, political support, and development potential. This paper logarithmizes the per capita GDP, the number of international Internet users, the number of patents

**Table 5. Variable declaration of spatial econometric model.**

| Variable | Abbreviation | Variable declaration | Data source |
|---|---|---|---|
| Economic condition | ECO | GDP per capita | statistical yearbook |
| Level of information technology | INF | Number of subscribers of Internet services per 10,000 people | statistical yearbook |
| Technological innovation capability | TEC | Patents applications granted | statistical yearbook |
| Government support | GOV | General public budget expenditure of transportation | statistical yearbook |
| Development potential of city | DEV | Growth rate of GDP | statistical yearbook |

granted, and the general public budget expenditure of the transportation industry to avoid the impact of heteroscedasticity. The variable description is shown in Table 5.

2. Model identification and test

In this paper, the SEM, SAR and SDM are constructed. Through LM test, LR test and Hausman test, the most suitable model for empirical analysis of the factors affecting logistics resilience is determined as the SDM. The test results are shown in Table 6.

Table 6 shows that the LM test results are all obvious, indicating that both the SEM model and the SAR model are applicable so the SDM model that combines the two models can be selected. Wald and LR tests are further conducted, and each statistic passes the significance test at the level of 1%, indicating that the SDM model will not degenerate into either the SEM model or SAR model. Hausman test results show that the fixed effect should be selected. The SDM model with fixed effect is superior based on the results of the aforementioned three tests.

## Analysis of factors influencing logistics resilience

SAR, SEM and SDM models are constructed respectively, and the estimation results are shown in Table 7.

According to the estimation results in Table 7, the SDM model has a higher degree of fitting and explanatory power. The SDM model with time fixed has a better fitting degree and stability when compared to the other SDM models with individual fixed and space-time double fixed. Therefore, the SDM model with time fixed effect is chosen in this study to analyze the influencing factors of urban logistics resilience and its spatial spillover effect in the Yangtze River Economic Belt.

Table 7 demonstrates that the spatial self-regression coefficient of the SDM model is 0.145, indicating that the urban logistics resilience in the Yangtze River Economic Belt has an obvious positive spillover effect. More specifically, every 1 percent increase in the logistics resilience of the city will result in a 0.145 percent increase in the logistics resilience of the surrounding cities.

**Table 6. Test results of spatial econometric model.**

| Test | Result | P value |
|---|---|---|
| LM-lag | 19.809 | 0.000 |
| Robust-LM-lag | 2.351 | 0.025 |
| LM-err | 40.192 | 0.000 |
| Robust-LM-err | 22.733 | 0.000 |
| LR-SDM→SEM = LR-err | 25 | 0.000 |
| LR-SDM→SAR = LR-lag | 27.37 | 0.000 |
| Wald-lag | 23.94 | 0.000 |
| Wald-error | 25.51 | 0.000 |
| Hausman | 24.55 | 0.000 |

**Table 7. Estimation results of three spatial econometric models.**

| Variable | SAR | | SEM | | SDM | |
|---|---|---|---|---|---|---|
| | coefficient | Z value | coefficient | Z value | coefficient | Z value |
| lnECO | 0.055*** | 5.29 | 0.053*** | 5.09 | 0.064*** | 6.08 |
| lnINF | 0.087*** | 10.89 | 0.088*** | 10.99 | 0.081*** | 10.8 |
| lnTEC | 0.006** | 2.31 | 0.006** | 1.38 | 0.006** | 2.35 |
| lnGOV | 0.001 | 1.76 | 0.002 | 0.98 | 0.001* | 1.65 |
| DEV | 0.001 | 1.88 | 0.000 | 0.73 | 0.001 | 0.61 |
| W×lnECO | | | | | 0.059*** | 2.64 |
| W×lnINF | | | | | -0.046** | -2.3 |
| W×lnTEC | | | | | -0.004 | -0.4 |
| W×lnGOV | | | | | -0.015*** | -3.05 |
| W×DEV | | | | | 0.005** | 2.02 |
| ρ(W×LC) | 0.038* | 1.64 | | | 0.145** | 1.92 |
| λ(W×ε) | | | 0.127** | 1.68 | | |
| R² | 0.432 | | 0.435 | | 0.503 | |
| Log-likelihood | 565.394 | | 566.582 | | 579.08 | |

Note

* * * means P < 0.01

* * means P < 0.05

* means P < 0.1, the same below.

In view of the different influencing factors of logistics resilience, the economic condition and the level of information technology play an important role in promoting the logistics resilience. The economic condition is the basis for the construction and development of regional logistics, and the improvement of informational level not only weakens the asymmetry of market information, but also promotes the industrial penetration and industrial integration of logistics with other industries, such as the integration of logistics industry, manufacturing industry and financial industry, which enriches the development path of regional logistics. Cities can improve the quality and operation of logistics services through the sharing of logistics resources and information, so as to improve the urban logistics resilience and sustainable development in the Yangtze River Economic Belt.

Technological innovation capability has a positive effect on the improvement of logistics resilience, and passed the 5% level of significance test. Due to the high industrial preference dependence of logistics industry, technological innovation not only promotes the development of manufacturing industry, but also improves the service quality and operation level of logistics industry. In addition, high-level technological innovation capability not only enhances the resilience of urban logistics, but also is an important support for achieving sustainable development goals. Scientific and technological innovation helps to control the energy consumption and waste emissions in packaging, transportation, warehousing and other aspects of logistics activities, so as to significantly improve the resource utilization and economic benefits of logistics enterprises, promote the construction of smart logistics and green logistics, and realize the sustainable development of regional logistics and economy.

Government support has improved the competitiveness of urban logistics to some extent, but its effect is limited. Logistics infrastructure such as roads and railways plays a decisive role in the operation of modern logistics. Logistics equipment such as logistics parks and logistics hubs also need to rely on logistics infrastructure to maintain normal operation. Although government support can improve the level of logistics infrastructure construction and operational

efficiency, enhance the liquidity of logistics resources and elements, but the resulting benefits have a certain lag, so the role of government support in promoting logistics resilience is limited. At the same time, environmental policies such as energy conservation and emission reduction promulgated by the government will also affect the normal operation of logistics enterprises and the adjustment of future development strategies, which will offset the support from government for the improvement of logistics resilience to a certain extent. The role of government in urban sustainable development needs support from all sectors of society such as enterprises, residents and public welfare organizations. Individual government guidance is difficult to promote the sustainable development. Therefore, the role of government support in promoting logistics resilience is limited.

The development potential of cities has no significant influence on the logistics resilience. The reason is that there is huge difference in the resource endowment, economic level and development foundation among the cities in the upper, middle and lower reaches of the Yangtze River Economic Belt. Although cities have great development potential, it still takes some time to improve their logistics resilience. Meanwhile, the logistics resilience is affected by many factors such as economy, society, science and technology. The development status and prospects of different cities in different fields are different. The promotion or inhibition of different factors will offset each other to a certain extent. Therefore, the calculation results of the impact of urban development potential on logistics resilience are not obvious.

## Spatial spillover effect analysis

The regression coefficient calculated above is only valid in the direction of action and the level of significance [13,76] because the SDM model includes the spatial lag term of the independent variable, which prevents the indirect effect of the independent variable on the dependent variable from being expressed solely by the regression coefficient. This study uses the partial differential method to unbiasedly process the estimated results of the SDM model, and then decomposes them into direct effect, indirect effect (i.e., spatial spillover effect), and total effect in order to more precisely measure the mechanism of various factors on logistics resilience. The estimated results of the above three effects are shown in Table 8.

According to Table 8, the economic condition has a direct effect coefficient of 0.166 and an indirect effect coefficient of 0.181, respectively. Additionally, they were successful in passing the significance test at the level of 1%. The economic condition is the explanatory variable that has the greatest impact on logistics resilience. It demonstrates how the foundation and impetus for the growth of the logistics sector is always the economy. As a result, the city and its neighboring cities will benefit greatly from an improvement in the economy.

The direct effect and indirect effect coefficient of informatization level are 0.080 and -0.040 respectively. They have passed the significance test at the level of 1% and 5% respectively. The result indicates that the improvement of informatization level has a significant pulling effect

**Table 8. Effect decomposition based on SDM model.**

| Variable | direct effect | | indirect effect | | total effect | |
|---|---|---|---|---|---|---|
| | coefficient | standard error | coefficient | standard error | coefficient | standard error |
| lnECO | 0.166*** | 0.010 | 0.181*** | 0.027 | 0.347*** | 0.030 |
| lnINF | 0.080*** | 0.008 | -0.040** | 0.022 | 0.040* | 0.025 |
| lnTEC | 0.017** | 0.005 | -0.015* | 0.012 | 0.032* | 0.013 |
| lnGOV | 0.021* | 0.002 | -0.016*** | 0.006 | 0.005*** | 0.006 |
| DEV | 0.001* | 0.001 | 0.006** | 0.003 | 0.007** | 0.003 |

on the logistics resilience of the city, but it will inhibit the logistics resilience of adjacent cities. The possible reason is that although information construction can promote information sharing and interconnection among cities, the factors such as talents, technology and capital required for information construction are competitive among cities. The differences in the development basis of different cities will also produce a certain siphon effect. Especially, the excessive attraction of central cities such as Shanghai, Hangzhou and Wuhan to various types of information and resources will form a Matthew effect, and gradually form a competitive situation of "the strong constant strong", which will restrict the improvement of the logistics resilience of surrounding cities.

The coefficients of direct effect and indirect effect of technological innovation capability are 0.017 and − 0.015, respectively. They have passed the significance test at the level of 5% and 10% respectively. The result indicates that technological innovation helps to improve the logistics resilience of the city, but there is a negative spillover effect on adjacent cities. The reason is that technological innovation helps to promote the promotion of multiple formats such as automatic distribution and reverse logistics, so as to promote the automation and informatization of the regional logistics, and thus enhance the logistics resilience of cities. However, the technological progress of the city will attract the inflow of talent, demand and other factors in the surrounding cities, thereby weakening the construction and development foundation of the logistics in the surrounding cities and inhibiting the improvement of their logistics resilience.

The direct and indirect effect coefficients of government support were 0.021 and -0.016, which passed the significance test of 10% and 1%, respectively. The result indicates that government support can promote the logistics resilience of the city, but it will have an obvious inhibitory effect on the logistics resilience of the surrounding cities. The reason is that the improvement of the urban transportation construction is helpful to enhance its logistics resilience, so as to absorb and undertake more logistics demand. However, it will also intensify the competition in the regional logistics market, especially between the central city and the edge city. For example, Chengdu has become the freight hub of Sichuan Province and even the southwest region by its dense traffic network and strong transport capacity. The growing logistics demand in turn also leads to the continuous increase in its transportation expenditure, and ultimately forms the Matthew effect, which makes it difficult for other cities in Sichuan Province to improve their logistics resilience.

The direct and indirect effects of urban development potential are significantly positive, and the coefficients are 0.001 and 0.006, respectively. It shows that cities can not only rely on economic development to enhance their logistics resilience, but also promote the logistics resilience of surrounding cities. On the one hand, the construction and development of emerging cities mean that they interact more closely and frequently with the outside world, which is conducive to revitalizing the logistics resources and information in the region and thus promoting inter-city freight exchanges. On the other hand, the rise of emerging cities can weaken the control and monopoly of core cities on logistics resources and information to a certain extent, and improve the liquidity of resource factors, so as to better play its spillover effect on the improvement of logistics resilience of surrounding cities.

## Robustness test

In order to verify the rationality and reliability of the research results, this paper uses the method of transforming spatial weight matrix to carry out robustness test. The geographical distance space weight matrix was constructed for the test, and the results are shown in Table 9.

Table 9 shows that the fitting degree of the SDM model is still the highest. Although the regression coefficients of the explanatory variables and their spatial lag terms have changed,

**Table 9. Robustness test results.**

| Variable | SAR | | SEM | | SDM | |
|---|---|---|---|---|---|---|
| | coefficient | Z value | coefficient | Z value | coefficient | Z value |
| lnECO | 0.057*** | 5.53 | 0.057*** | 5.46 | 0.076*** | 7.06 |
| lnINF | 0.085*** | 10.69 | 0.087*** | 10.60 | 0.077*** | 9.31 |
| lnTEC | 0.008* | 1.71 | 0.006* | 1.35 | 0.007** | 2.41 |
| lnGOV | 0.001 | 0.46 | 0.002 | 0.69 | 0.001* | 1.42 |
| DEV | 0.001 | 0.78 | 0.001 | 0.99 | 0.001 | 0.53 |
| W×lnECO | | | | | 0.558*** | 3.99 |
| W×lnINF | | | | | -0.458*** | -5.12 |
| W×lnTEC | | | | | 0.026 | 0.94 |
| W×lnGOV | | | | | -0.014*** | -1.39 |
| W×DEV | | | | | 0.005** | 1.88 |
| ρ(W×LC) | 0.055* | 2.36 | | | 0.082** | 1.5 |
| λ(W×ε) | | | -0.183* | -0.61 | | |
| R² | 0.471 | | 0.436 | | 0.533 | |
| Log-likelihood | 568.0954 | | 565.3852 | | 582.2872 | |

the positive and negative coefficients and the significance of the SDM model are basically consistent with the previous calculation results, indicating that the results of this study have strong robustness.

## Research findings

An extensive system for assessing logistics resilience in the Yangtze River Economic Belt was built. Taking cities as the basic research unit, the urban logistics resilience from 2016 to 2020 was calculated using the entropy TOPSIS method. The SDM model is built to empirically analyze the influencing factors of logistics resilience and its spatial spillover effect based on the analysis of the spatio-temporal evolution characteristics of logistics resilience. The main findings are as follows:

1. The logistics resilience of cities in the upper, middle, and lower reaches of the Yangtze River varies significantly. Cities in the lower reaches of the Yangtze River have high level of overall logistics resilience, while most cities in the middle and upper reaches of the Yangtze River have low levels of logistics industry construction and development. In addition to Shanghai and Chongqing, other cities affected by the outbreak see a sharp decline in their logistical competitiveness in 2020, and the overall level is significantly lower than it was before the outbreak.

2. There is an obvious spatial spillover effect of urban logistics resilience in the Yangtze River Economic Belt. The spatial spillover effect of urban logistics resilience is 0.145, that is, the improvement of logistics resilience of adjacent cities is conducive to the development of urban and regional logistics, and the improvement of logistics resilience will also have a positive influence on the logistics resilience of adjacent cities.

3. The improvement of economic condition, informatization level and technological innovation capability, government support and strong urban development potential will be all conducive to the logistics resilience of the city. The improvement of economic condition and development potential of a city can improve the logistics resilience of adjacent cities. However, due to the existence of "siphon effect" and "Matthew effect", the development of

information level, the enhancement of technological innovation capability and government support will hinder the improvement of the logistics resilience of adjacent cities.

## Discussion

### Innovations

In this paper, an extensive system for assessing logistics resilience in the Yangtze River Economic Belt was built and the urban logistics resilience from 2016 to 2020 was calculated. The SDM model is built to analyze the influencing factors of logistics resilience and its spatial spillover effect. Compared with similar studies, the innovations of this paper are described as below.

Firstly, different from the articles analyzing large-scale resilient systems such as ecological resilience [77] and economic resilience [78], this paper chooses logistics as the entry point to explore the impact of logistics resilience on the sustainable development of the Yangtze River Economic Belt. When measuring logistics resilience, in addition to considering economic, industrial and other factors, this study integrates the impact of logistics industry on the ecological environment into the logistics resilience assessment system from the perspective of sustainability, highlighting the importance of ecological environment on industrial and economic development.

Secondly, some studies use the Fuzzy-topsis [79] approach or Fuzzy-set Qualitative Comparative Analysis (fsQCA) [80] to calculate the influencing factors of logistics resilience, without considering its spillover effect. This paper considers the spatial factors, analyzes the impact of economic and technological level, innovation ability, policy support and development potential on the logistics resilience of the city and its surrounding cities, which reflects the realistic background of the coordinated development of the Yangtze River Economic Belt.

Thirdly, rather than using statistical data alone, this study selects both statistical data and POI data to evaluate logistics resilience. The application of POI data provides more possibilities for the study of resilience [81]. POI data can helps us identify the spatial distribution of different elements in the city. Improving the diversity of data sources and the flexibility of applications will help to promote resilience studies closer to the urban logistic construction and development needs.

### Suggestions

It is important to pay more attention to logistics resilience and make full use of the advantages of the Yangtze River as a "golden waterway" to develop multimodal transport. The initiative and adaptability of regional logistics system can be enhanced through establishing and improving the logistics emergency support system and emergency linkage mechanism.

In order to promote the sustainable development of Yangtze River Economic Belt, it is significant to use the spatial spillover effect of logistics resilience and strengthen the ties and cooperation between cities in the different classes. For example, core cities such as Shanghai, Suzhou and Ningbo in the lower reaches of the Yangtze River have high logistics resilience, so it is necessary to focus on the construction of national logistics center cities in the Yangtze River Delta City cluster, and promote the development of regional logistics through "agglomeration to re-agglomeration" of logistics elements. The leading role of Wuhan and Changsha in the middle reaches is weak. On the one hand, these two cities can promote the agglomeration of logistics industry through their own advantageous industrial clusters such as intelligent manufacturing. On the other hand, the logistics industry construction of cities with advantages in transportation such as Ganzhou and Jian should also be paid attention to. Chongqing and

Chengdu, two logistics centers in the upper reaches of the Yangtze River, have produced "crowding out effect" on the development of logistics in the surrounding cities. The logistics elements are further concentrated in Chongqing and Chengdu, which aggravate the unbalanced and uncoordinated development of logistics in Guiyang, Kunming and other cities in the upper reaches. Therefore, the upper reaches of the Yangtze River should improve the intercity transportation network, focus on strengthening the logistics infrastructure construction in some remote cities, and expand the radiation range of Chongqing, Chengdu and other logistics center cities, so as to realize the mutual promotion and coordinated development of the logistics industry among upstream cities.

The improvement of informatization level and technological innovation ability is conducive to improving the resilience of urban logistics. Therefore, technological innovation should be highlighted, and the levels of urban logistics information and innovation ability are suggested to be comprehensively improved. The application of advanced technologies such as Internet, artificial intelligence and blockchain in the process of logistics warehousing, distribution and recycling should be strengthened. For example, through the construction of intelligent transportation system, logistics transportation and distribution routes could be reasonably planned to reduce energy consumption and exhaust emissions in the transportation process. It is also significant to establish a reverse logistics system complementary to forward logistics combined with Internet technology, which can realize the recycling of resources. Promoting the construction and development of green logistics and intelligent logistics can help to promote the sustainable development of economic and ecological benefits of logistics enterprises.

Government intervention can help improve urban logistics resilience. Therefore, various existing policy barriers should be removed. Cooperation and coordination between different cities should be strengthened from micro affairs and specific projects, such as the integration of port and station construction and the optimization of regional collaborative distribution network, so as to achieve the purpose of enhancing the local ability to integrate innovation, capital, information and other resources, and improving the overall regional logistics resilience.

The "siphon effect" and "Matthew effect" should be avoided as far as possible. At the macro-level, it is important to strengthen the construction of intermodal transportation between the three major shipping centers of Chongqing, Wuhan, Shanghai and other cities, give full play to the advantages of the Yangtze River as a "golden waterway" in waterway transportation and resource allocation, and realize the clustering of the Yangtze River waterway ports. At the micro-aspect, government support has a negative impact on logistics resilience of adjacent cities because of the "siphon effect", therefore, cities such as Lijiang, Puer and Zhangjiajie should identify the shortcomings of their own logistics industry construction, reduce the gap between the core cities through cooperation and assistance, and promote the balanced development of urban logistics.

Finally, the Yangtze Economic Belt connects the Yangtze River Delta, China's richest region, with the less developed upper reaches of the river. Due to the positive spillover effect of urban logistics resilience, economic development and urban development potential, cities in the upstream area limited by natural conditions and other factors can realize "driving the weak with the strong" and promote the development of surrounding cities by strengthening the infrastructure construction and enhancing their development capacity.

## Conclusions

From the micro perspective of cities, this paper analyzes the spatial and temporal evolution characteristics, influencing factors and spatial spillover effects of logistics resilience of Yangtze

River Economic Belt, and explores the realization path of sustainable development based on the perspective from logistics resilience. However, this study also has some limitations. On the one hand, due to data acquisition limitations, this paper only calculates the logistics resilience of 110 prefecture-level cities in the Yangtze River Economic Belt, without considering the logistics resilience of ethnic minority autonomous prefectures located in the middle and upper reaches of the Yangtze River. In fact, affected by natural conditions, traffic conditions and other factors, the development of logistics industry in these areas are more slowly. Therefore, in the future, it is considered to analyze the logistics resilience of ethnic minority autonomous prefectures through field investigation, expanding the scope of data search and other methods, so as to obtain more targeted and feasible development suggestions. On the other hand, the analysis of logistics resilience and sustainable development path of the Yangtze River Economic Belt is still at the municipal level. In the future, the research unit can be reduced to the county level, or the analysis results of different scales can be compared, so as to put forward more detailed and targeted suggestions for the improvement of logistics toughness and sustainable development of the Yangtze River Economic Belt.

At the same time, quantitative methods such as network analysis [82,83] and differential game [84] also provides more possibilities for the study of resilience. The network structure determines the network function, and the network topology can reflect the stability of the network. By calculating the relevant indicators of the topology, the resilience of the logistics network can be characterized from the perspective of structural stability and network game. In addition, expanding the time section and comparing the similarities and differences between the influencing factors of urban logistics resilience before and after the epidemic and the spatial spillover effects are helpful to understand the impact of the COVID-19 epidemic on the logistics industry, so as to obtain more targeted analysis results to respond to similar events in the future.

## Author Contributions

**Conceptualization:** Xiaofan Zhang, Yin Huang.

**Data curation:** Xiaofan Zhang, Yin Huang.

**Formal analysis:** Yin Huang.

**Funding acquisition:** Xiaofan Zhang, Yin Huang.

**Methodology:** Xiaofan Zhang, Yin Huang.

**Software:** Xiaofan Zhang.

**Supervision:** Yin Huang.

**Validation:** Yin Huang.

**Visualization:** Xiaofan Zhang.

**Writing – original draft:** Xiaofan Zhang.

**Writing – review & editing:** Xiaofan Zhang, Yin Huang.

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
