## [Decision Letter · Decision Letter 0]

18 Sep 2023

PONE-D-23-16786Spatial evolution, influencing factors and spillover effects of logistics resilience in the Yangtze River Economic BeltPLOS ONE

Dear Dr. Huang,

Thank you for submitting your manuscript to PLOS ONE. After careful consideration, we feel that it has merit but does not fully meet PLOS ONE’s publication criteria as it currently stands. Therefore, we invite you to submit a revised version of the manuscript that addresses the points raised during the review process.

This paper is interesting and find some valuable conclusion，the authors evaluated urban logistics resilience combined with POI data and statistical data. Supply-demand relationship, industrial structure and ecological environment were considered in the process of calculation of logistics resilience. In addition, a spatial Durbin model was constructed to reveal the influencing factors of logistics resilience and the interaction effect between cities. However, the paper can be accepted after making the following minor revisions.

1. The innovation of this paper needs to be highlighted in the abstract. The abstract need to rewrite.

2. The authors need to add the background of this research.

3. The literature review is not enough, the innovation of this paper and the contribution made by previous studies have not been clearly expressed. The author should summarize the existing research gaps and highlight the innovation of this paper after completing the literature review.

4. It is suggested that the authors add more descriptions of the reasons for choosing Yangtze River Economic Belt as a case study, the following literature should be helpful for your research：(1) Adaptability analysis of water pollution and advanced industrial structure in Jiangsu Province, China. (2)Coordination of the Industrial-Ecological Economy in the Yangtze River Economic Belt, China. (3)Compilation of Water Resource Balance Sheets under Unified Accounting of Water Quantity and Quality, a Case Study of Hubei Province.

5. The author needs to explain the meaning of logistics resilience and explain the reasons why it includes four aspects: logistics supply, logistics demand, industrial structure, and impact on environment

We look forward to receiving your revised manuscript.

Kind regards,

Fuyou Guo, (Ph.D.

Academic Editor

PLOS ONE

Journal Requirements:

"Yin Huang:this study was supported by the National Natural Science Foundation of China (Grant no.72174214); Key Project of Scientific Research of Department of Education of Hunan Province (Grant no. 21A0156).

Xiaofan Zhang: this study was supported by Hunan Provincial Innovation Foundation for Postgraduate (Grant no. CX20220737)." 

4. We note that Figures 1, 2, 3, and 4 in your submission contain [map/satellite] images which may be copyrighted. All PLOS content is published under the Creative Commons Attribution License (CC BY 4.0), which means that the manuscript, images, and Supporting Information files will be freely available online, and any third party is permitted to access, download, copy, distribute, and use these materials in any way, even commercially, with proper attribution. For these reasons, we cannot publish previously copyrighted maps or satellite images created using proprietary data, such as Google software (Google Maps, Street View, and Earth). For more information, see our copyright guidelines: http://journals.plos.org/plosone/s/licenses-and-copyright.

1.You may seek permission from the original copyright holder of Figures 1, 2, 3, and 4 to publish the content specifically under the CC BY 4.0 license. 

5. Please include captions for your Supporting Information files at the end of your manuscript, and update any in-text citations to match accordingly. Please see our Supporting Information guidelines for more information: http://journals.plos.org/plosone/s/supporting-information

Additional Editor Comments :

This paper is interesting and find some valuable conclusion，the authors evaluated urban logistics resilience combined with POI data and statistical data. Supply-demand relationship, industrial structure and ecological environment were considered in the process of calculation of logistics resilience. In addition, a spatial Durbin model was constructed to reveal the influencing factors of logistics resilience and the interaction effect between cities. However, the paper can be accepted after making the following minor revisions.

1. The innovation of this paper needs to be highlighted in the abstract. The abstract need to rewrite.

2. The authors need to add the background of this research.

3. The literature review is not enough, the innovation of this paper and the contribution made by previous studies have not been clearly expressed. The author should summarize the existing research gaps and highlight the innovation of this paper after completing the literature review.

4. It is suggested that the authors add more descriptions of the reasons for choosing Yangtze River Economic Belt as a case study, the following literature should be helpful for your research：(1) Adaptability analysis of water pollution and advanced industrial structure in Jiangsu Province, China. (2)Coordination of the Industrial-Ecological Economy in the Yangtze River Economic Belt, China. (3)Compilation of Water Resource Balance Sheets under Unified Accounting of Water Quantity and Quality, a Case Study of Hubei Province.

5. The author needs to explain the meaning of logistics resilience and explain the reasons why it includes four aspects: logistics supply, logistics demand, industrial structure, and impact on environment

Reviewers' comments:

Reviewer's Responses to Questions

**Comments to the Author**

1. Is the manuscript technically sound, and do the data support the conclusions?

Reviewer #1: Yes

2. Has the statistical analysis been performed appropriately and rigorously? 

Reviewer #1: Yes

3. Have the authors made all data underlying the findings in their manuscript fully available?

Reviewer #1: Yes

4. Is the manuscript presented in an intelligible fashion and written in standard English?

Reviewer #1: Yes

5. Review Comments to the Author

Reviewer #1: This paper is interesting and find some valuable conclusion，the authors evaluated urban logistics resilience combined with POI data and statistical data. Supply-demand relationship, industrial structure and ecological environment were considered in the process of calculation of logistics resilience. In addition, a spatial Durbin model was constructed to reveal the influencing factors of logistics resilience and the interaction effect between cities. However, the paper can be accepted after making the following minor revisions.

1. The innovation of this paper needs to be highlighted in the abstract. The abstract need to rewrite.

2. The authors need to add the background of this research.

3. The literature review is not enough, the innovation of this paper and the contribution made by previous studies have not been clearly expressed. The author should summarize the existing research gaps and highlight the innovation of this paper after completing the literature review.

4. It is suggested that the authors add more descriptions of the reasons for choosing Yangtze River Economic Belt as a case study, the following literature should be helpful for your research：(1) Adaptability analysis of water pollution and advanced industrial structure in Jiangsu Province, China. (2)Coordination of the Industrial-Ecological Economy in the Yangtze River Economic Belt, China. (3)Compilation of Water Resource Balance Sheets under Unified Accounting of Water Quantity and Quality, a Case Study of Hubei Province.

5. The author needs to explain the meaning of logistics resilience and explain the reasons why it includes four aspects: logistics supply, logistics demand, industrial structure, and impact on environment

6. PLOS authors have the option to publish the peer review history of their article (what does this mean?). If published, this will include your full peer review and any attached files.

Reviewer #1: No

---

## [Author Response · Author response to Decision Letter 0]

12 Feb 2024

Thank you for your letter and for the reviewer/editorial comments concerning our manuscript entitled “Spatial evolution, influencing factors and spillover effects of logistics resilience in the Yangtze River Economic Belt” (ID:PONE-D-23-16768).

Those comments are all valuable and very helpful for revising and improving our paper, as well as the important guiding significance to our research. We have studied comments carefully and have made correction which we hope will meet with your approval. All revised portions are marked in red, and the deleted parts have been marked in the track changes mode in the revised manuscript which we would like to submit for your kind consideration.

1. The innovation of this paper needs to be highlighted in the abstract. The abstract need to rewrite.

Response:

Thanks for your suggestion. We have rewritten the abstract, mainly expounding the importance of studying logistics resilience and the innovation points of this paper in the research perspective, logistics resilience evaluation system design and data selection. Due to the word limit of the abstract, we introduce the innovation of the research perspective, research method and data selection in detail in the discussion part of the paper. First, based on the perspective of economic geography, a spatial econometric model is constructed to analyze the influencing factors that lead to the differences in logistics resilience of different cities and the existing spatial spillover effect. Second, in the design of logistics resilience rating system, considering the market, external environment and ecology, the urban logistics resilience is evaluated from the supply and demand relationship, industrial structure and the impact on the environment. Third, in terms of data selection, this paper comprehensively selects statistical data and POI data into the model for calculation, and makes an innovative attempt in data selection. Please see the abstract in the revised manuscript.

2. The authors need to add the background of this research.

Response:

Thank you for your proposal on adding the background of this research. First, we revised the introduction, emphasizing that the research background of this paper is the post-COVID-19 era, explaining the important role of the logistics industry and logistics resilience in economic recovery after the COVID-19 pandemic, and introducing the geographical location of the Yangtze River Economic Belt. Second, we have made major revisions to the literature review. In part 2.2 "Study on logistics industry and logistics resilience in the Yangtze River Economic Belt", the development status of regional development imbalance in the Yangtze River Economic Belt is introduced, the important role of regional logistics construction for high-quality and sustainable development of regional economy is expounded, and the shortcomings of previous studies on logistics resilience in the Yangtze River Economic Belt are summarized. Please see the introduction and literature review in the revised manuscript.

3. The literature review is not enough, the innovation of this paper and the contribution made by previous studies have not been clearly expressed. The author should summarize the existing research gaps and highlight the innovation of this paper after completing the literature review.

Response:

Thank you for pointing out our mistakes. We have made a major revision to the literature review, which is divided into three parts. 

(1)In section 2.1 "Study on resilience and logistics resilience", we sorted out the definition of the word "resilience" and the process of concept expansion, summarized the relevant definitions of supply chain resilience and logistics resilience, and presented them in Table 1 of the revised manuscript. By reviewing and summarizing the research on supply chain resilience and logistics resilience, we define logistics resilience. 

(2)We changed the second part of the literature review to "Study on logistics industry and logistics resilience in the Yangtze River Economic Belt", reviewed relevant research on the development of the logistics industry in the Yangtze River Economic Belt, and highlighted the important role of the logistics industry in the high-quality and sustainable development of the Yangtze River Economic Belt.

(3) We summarized the shortcomings of the existing studies on the logistics resilience of the Yangtze River Economic Belt, and introduce the main research content and innovation points of this paper. Please see the literature review in the revised manuscript.

4. It is suggested that the authors add more descriptions of the reasons for choosing Yangtze River Economic Belt as a case study, the following literature should be helpful for your research：(1) Adaptability analysis of water pollution and advanced industrial structure in Jiangsu Province, China. (2)Coordination of the Industrial-Ecological Economy in the Yangtze River Economic Belt, China. (3)Compilation of Water Resource Balance Sheets under Unified Accounting of Water Quantity and Quality, a Case Study of Hubei Province.

Response:

Thanks for your suggestion on adding more descriptions of the reasons for choosing Yangtze River Economic Belt as a case study. Referring to the several literature you listed, we have added descriptions of the development status of the Yangtze River Economic Belt in the introduction and literature review, which is also the reason why this paper chooses the Yangtze River Economic Belt as the study region. Please see the references in the revised manuscript.

5.The author needs to explain the meaning of logistics resilience and explain the reasons why it includes four aspects: logistics supply, logistics demand, industrial structure, and impact on environment.

Response:

We agree with the reviewer’s opinions on explaining the meaning of logistics resilience and the design idea of logistics resilience evaluation system. First, in the part of literature review, we defined the concept of logistics resilience by reviewing the existing studies on supply chain resilience and logistics resilience. Logistics resilience is defined as the resistance of the regional logistics system when it is disturbed by external economic, natural and other environmental conditions, or the recovery and improvement ability of the regional logistics system to a better path through system optimization. Second, in the method part, it explains the reasons and background of choosing these four angles to measure logistics resilience: (1) Logistics supply and logistics demand: logistics supply and demand have a profound impact on the performance of the logistics market, which further affects the resilience of logistics. (2) Industrial structure: as a service industry, logistics is closely related to other industries, especially manufacturing, so changes in industrial structure will also affect logistics resilience by affecting logistics demand. (3) Impact on environment: transportation activities in the logistics industry are the main source of environmental pollution, and their impact on the ecological environment will eventually affect the sustainable development of the logistics industry through the feedback mechanism. Therefore, paying attention to the impact of the development of the logistics industry on the ecological environment is of great significance for the evaluation and improvement of logistics resilience. Please see the literature review and methodology in the revised manuscript.

---

## [Decision Letter · Decision Letter 1]

27 Feb 2024

PONE-D-23-16786R1Spatial evolution, influencing factors and spillover effects of logistics resilience in the Yangtze River Economic BeltPLOS ONE

Dear Dr. Huang,

Thank you for submitting your manuscript to PLOS ONE. After careful consideration, we feel that it has merit but does not fully meet PLOS ONE’s publication criteria as it currently stands. Therefore, we invite you to submit a revised version of the manuscript that addresses the points raised during the review process.

(1) In the "Abstract" section, it is better to describe the research by mentioning the quantitative results of study. Readers should be able to see the quantities of results in the abstract. the abstract should also contain Objectives, Methods/Analysis, Findings, and Novelty /Improvement

(2) In the "Introduction" section, the introduction expounds too much very general and well-known information, and the importance and significance of the research are not highlighted

(3) In the "Introduction" section, The author lacks references when expounding the research background.

(4) For readers to quickly catch your contribution, it would be better to highlight major difficulties and challenges, and your original achievements to overcome them, in a clearer way in abstract and introduction

(5) More explanation is needed for where there is a research gap and what the goals of the research are. The research gap and the goals of the research are not explained in detail which leads to the reader missing the significance of the research.

(6) In the second section, Please provide a framework diagram of this paper

(7) The analysis of the results is very superficial. It simply describes the change trend of the data, and lacks in-depth analysis and interpretation.

(8) How can your study inspire studies on other regions? The results should be further elaborated to show how they could be used for the real applications

(9) The article lacks an important discussion link, in which the author should focus on describing the differences between the article study and other scholars' studies, thus highlighting the relevance and academic value of the article, the following literature should be helpful for your research: (1)Reduction pathways identification of Agricultural Water Pollution in Hubei Province, China. (2) A differential game of water pollution management in the trans-jurisdictional river basin\\

(10) The article was not written following the correct journal's guidelines to be considered for publication. INTORDCUTION→MRTHOD→RESULTS→DISSCUSION→CONCLUSION

We look forward to receiving your revised manuscript.

Kind regards,

Fuyou Guo, (Ph.D.

Academic Editor

PLOS ONE

Journal Requirements:

Additional Editor Comments:

(1) In the "Abstract" section, it is better to describe the research by mentioning the quantitative results of study. Readers should be able to see the quantities of results in the abstract. the abstract should also contain Objectives, Methods/Analysis, Findings, and Novelty /Improvement

(2) In the "Introduction" section, the introduction expounds too much very general and well-known information, and the importance and significance of the research are not highlighted

(3) In the "Introduction" section, The author lacks references when expounding the research background.

(4) For readers to quickly catch your contribution, it would be better to highlight major difficulties and challenges, and your original achievements to overcome them, in a clearer way in abstract and introduction

(5) More explanation is needed for where there is a research gap and what the goals of the research are. The research gap and the goals of the research are not explained in detail which leads to the reader missing the significance of the research.

(6) In the second section, Please provide a framework diagram of this paper

(7) The analysis of the results is very superficial. It simply describes the change trend of the data, and lacks in-depth analysis and interpretation.

(8) How can your study inspire studies on other regions? The results should be further elaborated to show how they could be used for the real applications

(9) The article lacks an important discussion link, in which the author should focus on describing the differences between the article study and other scholars' studies, thus highlighting the relevance and academic value of the article, the following literature should be helpful for your research: (1)Reduction pathways identification of Agricultural Water Pollution in Hubei Province, China. (2) A differential game of water pollution management in the trans-jurisdictional river basin\\

(10) The article was not written following the correct journal's guidelines to be considered for publication. INTORDCUTION→MRTHOD→RESULTS→DISSCUSION→CONCLUSION

Reviewers' comments:

Reviewer's Responses to Questions

**Comments to the Author**

1. If the authors have adequately addressed your comments raised in a previous round of review and you feel that this manuscript is now acceptable for publication, you may indicate that here to bypass the “Comments to the Author” section, enter your conflict of interest statement in the “Confidential to Editor” section, and submit your "Accept" recommendation.

Reviewer #1: (No Response)

2. Is the manuscript technically sound, and do the data support the conclusions?

Reviewer #1: (No Response)

3. Has the statistical analysis been performed appropriately and rigorously? 

Reviewer #1: (No Response)

4. Have the authors made all data underlying the findings in their manuscript fully available?

Reviewer #1: (No Response)

5. Is the manuscript presented in an intelligible fashion and written in standard English?

Reviewer #1: (No Response)

6. Review Comments to the Author

Reviewer #1: (1) In the "Abstract" section, it is better to describe the research by mentioning the quantitative results of study. Readers should be able to see the quantities of results in the abstract. the abstract should also contain Objectives, Methods/Analysis, Findings, and Novelty /Improvement

(2) In the "Introduction" section, the introduction expounds too much very general and well-known information, and the importance and significance of the research are not highlighted

(3) In the "Introduction" section, The author lacks references when expounding the research background.

(4) For readers to quickly catch your contribution, it would be better to highlight major difficulties and challenges, and your original achievements to overcome them, in a clearer way in abstract and introduction

(5) More explanation is needed for where there is a research gap and what the goals of the research are. The research gap and the goals of the research are not explained in detail which leads to the reader missing the significance of the research.

(6) In the second section, Please provide a framework diagram of this paper

(7) The analysis of the results is very superficial. It simply describes the change trend of the data, and lacks in-depth analysis and interpretation.

(8) How can your study inspire studies on other regions? The results should be further elaborated to show how they could be used for the real applications

(9) The article lacks an important discussion link, in which the author should focus on describing the differences between the article study and other scholars' studies, thus highlighting the relevance and academic value of the article, the following literature should be helpful for your research: (1)Reduction pathways identification of Agricultural Water Pollution in Hubei Province, China. (2) A differential game of water pollution management in the trans-jurisdictional river basin\\

(10) The article was not written following the correct journal's guidelines to be considered for publication. INTORDCUTION→MRTHOD→RESULTS→DISSCUSION→CONCLUSION

7. PLOS authors have the option to publish the peer review history of their article (what does this mean?). If published, this will include your full peer review and any attached files.

Reviewer #1: No

---

## [Author Response · Author response to Decision Letter 1]

12 Apr 2024

1. In the "Abstract" section, it is better to describe the research by mentioning the quantitative results of study. Readers should be able to see the quantities of results in the abstract. the abstract should also contain Objectives, Methods/Analysis, Findings, and Novelty /Improvement.

Response:

Thanks for your suggestion. We have rewritten the abstract, mainly adding the quantitative results of this study and the possible directions for future improvement. Due to the word limit of the abstract, we introduce the innovation of the research perspective, research method and data selection in detail in the discussion part of the paper. First, based on the perspective of economic geography, a spatial econometric model is constructed to analyze the influencing factors that lead to the differences in logistics resilience of different cities and the existing spatial spillover effect. Second, in the design of logistics resilience rating system, considering the market, external environment and ecology, the urban logistics resilience is evaluated from the supply and demand relationship, industrial structure and the impact on the environment. Third, in terms of data selection, this paper comprehensively selects statistical data and POI data into the model for calculation, and makes an innovative attempt in data selection. Please see the abstract in the revised manuscript.

2. In the "Introduction" section, the introduction expounds too much very general and well-known information, and the importance and significance of the research are not highlighted.

Response:

Thank you for your valuable suggestion. The introduction explains the significance of this study from three perspectives: the importance of logistics industry to global economic recovery, the significance of logistics resilience in the process of regional sustainable development and the imbalance of logistics industry development in the Yangtze River Economic Belt. We have deleted part of the introduction to make it more concise and to better highlight the practical significance of conducting this research. Please see the introduction in the revised manuscript.

3.In the "Introduction" section, The author lacks references when expounding the research background.

Response:

Thank you for your proposal on adding more references in the “Introduction” section. We have removed parts of the introduction and made appropriate citations in existing references in the article. Please see the introduction in the revised manuscript.

4.For readers to quickly catch your contribution, it would be better to highlight major difficulties and challenges, and your original achievements to overcome them, in a clearer way in abstract and introduction.

Response:

Thanks for your suggestion. We have rewritten the abstract, mainly adding the quantitative results of this study and the possible directions for future improvement. At the same time, we also explain the data sources of this study in the abstract, corresponding to the innovation of this paper mentioned at the end of the paper. Please see the abstract in the revised manuscript.

5.More explanation is needed for where there is a research gap and what the goals of the research are. The research gap and the goals of the research are not explained in detail which leads to the reader missing the significance of the research.

Response:

We agree with the reviewer’s opinions on explaining the research gap and research goal of this study. In the literature review, by combing the relevant literature of resilience research, we summarized the gaps in current research and expounded the significance of carrying out this research. At the end of the literature review, we inserted a research framework diagram to explain the purpose and main contents of this study. Please see the summary of the literature review and the research framework in the revised manuscript.

6.In the second section, Please provide a framework diagram of this paper.

Response:

Thanks for your suggestion on adding a framework diagram of this paper. At the end of the literature review, we add the framework diagram of this paper. Please see Fig 1 in the revised manuscript.

7.The analysis of the results is very superficial. It simply describes the change trend of the data, and lacks in-depth analysis and interpretation.

Response:

Thanks for your suggestion on adding in-depth analysis and interpretation. First of all, In the part of logistics resilience measurement, we separately explain the calculation results of logistics resilience of several special cities by giving examples, and emphasize the change of logistics resilience of the Yangtze River Economic Belt over time to highlight the fine granularity of the study, so as to make our research results more typical. Second, the spatio-temporal distribution of logistics resilience is the foundation for subsequent research, rather than the main part of this study. We substitute the calculation results of logistics resilience into the subsequent spatial Durbin model as explained variables to calculate the influence and spillover effect of different factors on logistics resilience. Therefore, considering the word limit of the article, we did not give a very detailed describe the distribution characteristics of logistics resilience. Finally, in our original manuscript, we first described the calculated quantitative results in the part of impact factor analysis and spillover effect measurement, and then analyzed the possible reasons for the results. The relevant content of the result analysis is highlighted in red. Please see the results in the revised manuscript.

8.How can your study inspire studies on other regions? The results should be further elaborated to show how they could be used for the real applications.

Response:

Thanks for your suggestion on elaborating how the results could be used for the real applications. In the suggestion section, we discussed the aspects in which cities in the Yangtze River Economic Belt should strengthen construction to promote their own logistics resilience according to the influencing factors and spillover effect analysis results. As for the implications of this study for other regions, we believe that the Yangtze River Economic Belt connects developed and less developed regions, and it can be seen as a microcosm of a country, and explain how other regions can take the spatial spillover effect between cities in the Yangtze River Economic Belt to improve their own logistics resilience. Please see the suggestions in the revised manuscript.

9.The article lacks an important discussion link, in which the author should focus on describing the differences between the article study and other scholars' studies, thus highlighting the relevance and academic value of the article, the following literature should be helpful for your research: (1)Reduction pathways identification of Agricultural Water Pollution in Hubei Province, China. (2) A differential game of water pollution management in the trans-jurisdictional river basin\\.

Response:

Thanks for your suggestion. In the discussion section at the end of the paper, we put forward the innovation in the research perspective, research methods and data sources by comparing the existing researches. In the conclusion part, we summarized the limitations of this study and the direction for improvement in the future by comparing the studies of other scholars. We cited a reference you provided, which provided us with a feasible research method for future research. Please see discussion and conclusions in the revised manuscript.

10.The article was not written following the correct journal's guidelines to be considered for publication. INTORDCUTION→METHOD→RESULTS→DISSCUSION→CONCLUSION.

Response:

Thank you for your reminding. As for the literature review, by combing the relevant literature of resilience research, we found the gap of current research, which is the basis of this study. We adopted the comments of the first review, and added a summary of the existing studies in the literature review. Based on this, we explained the objectives of this study and highlighted the purpose and significance of this study. Therefore, we did not delete the literature review. We have changed the title of the second half of the article according to your suggestion.

---

## [Decision Letter · Decision Letter 2]

30 Apr 2024

Spatial evolution, influencing factors and spillover effects of logistics resilience in the Yangtze River Economic Belt

PONE-D-23-16786R2

Dear Dr. Huang,

We’re pleased to inform you that your manuscript has been judged scientifically suitable for publication and will be formally accepted for publication once it meets all outstanding technical requirements.

Kind regards,

Fuyou Guo, (Ph.D.

Academic Editor

PLOS ONE

Additional Editor Comments (optional):

Reviewers' comments:

Reviewer's Responses to Questions

**Comments to the Author**

1. If the authors have adequately addressed your comments raised in a previous round of review and you feel that this manuscript is now acceptable for publication, you may indicate that here to bypass the “Comments to the Author” section, enter your conflict of interest statement in the “Confidential to Editor” section, and submit your "Accept" recommendation.

Reviewer #1: All comments have been addressed

2. Is the manuscript technically sound, and do the data support the conclusions?

Reviewer #1: (No Response)

3. Has the statistical analysis been performed appropriately and rigorously? 

Reviewer #1: (No Response)

4. Have the authors made all data underlying the findings in their manuscript fully available?

Reviewer #1: (No Response)

5. Is the manuscript presented in an intelligible fashion and written in standard English?

Reviewer #1: (No Response)

6. Review Comments to the Author

Reviewer #1: It seems that all my comments had been addressed. The quality of the paper has been significantly enhanced. I think your work has reached the level suitable for publication, and I recommend it for publication.

7. PLOS authors have the option to publish the peer review history of their article (what does this mean?). If published, this will include your full peer review and any attached files.

Reviewer #1: No

---

## [Editor Report · Acceptance letter]

16 May 2024

PONE-D-23-16786R2 

PLOS ONE

Dear Dr. Huang, 

I'm pleased to inform you that your manuscript has been deemed suitable for publication in PLOS ONE. Congratulations! Your manuscript is now being handed over to our production team.

Kind regards, 

on behalf of

Associate professor Fuyou Guo 

Academic Editor

PLOS ONE